# The Reversion of the Epigenetic Signature of Coronary Heart Disease in Response to Smoking Cessation

**DOI:** 10.3390/genes14061233

**Published:** 2023-06-08

**Authors:** Robert Philibert, Joanna Moody, Willem Philibert, Meeshanthini V. Dogan, Eric A. Hoffman

**Affiliations:** 1Department of Psychiatry, University of Iowa, Iowa City, IA 52242, USA; joanna-moody@uiowa.edu (J.M.); willem-philibert@uiowa.edu (W.P.); 2Cardio Diagnostics Inc., Chicago, IL 60642, USA; mdogan@cardiodiagnosticsinc.com; 3Department of Biomedical Engineering, University of Iowa, Iowa City, IA 52242, USA; eric-hoffman@uiowa.edu; 4Department of Radiology, University of Iowa, Iowa City, IA 52242, USA

**Keywords:** coronary heart disease, treatment monitoring, DNA methylation, smoking cessation, epigenetics

## Abstract

Coronary heart disease (CHD) is the leading cause of death worldwide. However, current diagnostic tools for CHD, such as coronary computed tomography angiography (CCTA), are poorly suited for monitoring treatment response. Recently, we have introduced an artificial-intelligence-guided integrated genetic–epigenetic test for CHD whose core consists of six assays that determine methylation in pathways known to moderate the pathogenesis of CHD. However, whether methylation at these six loci is sufficiently dynamic to guide CHD treatment response is unknown. To test that hypothesis, we examined the relationship of changes in these six loci to changes in cg05575921, a generally accepted marker of smoking intensity, using DNA from a cohort of 39 subjects undergoing a 90-day smoking cessation intervention and methylation-sensitive digital PCR (MSdPCR). We found that changes in epigenetic smoking intensity were significantly associated with reversion of the CHD-associated methylation signature at five of the six MSdPCR predictor sites: cg03725309, cg12586707, cg04988978, cg17901584, and cg21161138. We conclude that methylation-based approaches could be a scalable method for assessing the clinical effectiveness of CHD interventions, and that further studies to understand the responsiveness of these epigenetic measures to other forms of CHD treatment are in order.

## 1. Introduction

Coronary heart disease (CHD) is the leading cause of death in the world, with over 9 million individuals dying annually from myocardial infarctions [1]. Up to 90% of these deaths may be preventable [2]. However, to achieve this goal, more scalable approaches for both recognizing and monitoring those at risk of CHD for initial preventive intervention and those with current CHD for more intensive therapy are necessary.

Unfortunately, with respect to the latter, there are significant limitations in utilizing contemporary CHD diagnostic approaches for monitoring the effectiveness of interventions, both lifestyle and therapeutic. The currently accepted methods for diagnosing stable CHD from the American Heart Association/American College of Cardiology include exercise stress tests with electrocardiograms (ECG), exercise or pharmacological stress tests with echocardiography, myocardial perfusion imaging, coronary computed tomographic angiography (CCTA), and coronary angiography [3]. The choice of which test is employed is dependent on a number of factors, including the degree of clinical suspicion for CHD, the ability to exercise, the likelihood of experiencing high-risk events, and the availability of testing resources [4]. Unfortunately, each of those modalities has significant limitations that affect their clinical implementation. For example, the exercise ECG is the most commonly used and least invasive method, but has only 58% sensitivity and 62% specificity [3]. Perfusion imaging has better sensitivity (~87%) and specificity (70%), but requires expensive infrastructure, specialized interpretation, and requires considerable radiation exposure (15–35 millisieverts) [5,6]. The final method is coronary angiography, which is considered by many to be the “gold standard” for the assessment of CHD. However, it is invasive, entails considerable exposure to radiation and iodinated contrast dyes, and is insensitive to non-obstructive forms of CHD. More importantly, these diagnostic tools fall short in providing actionable information to personalize patient care and lack the ability to measure the effectiveness of interventions. Hence, there is considerable need for less costly and invasive methods for assessing CHD status that also provide actionable information that can be leveraged by healthcare providers to tailor, refine, and enhance treatment plans for each patient more effectively.

Recently, Cardio Diagnostics has introduced an artificial intelligence-guided genetic–epigenetic test (PrecisionCHD™) for assessing current CHD [7]. The test uses a machine learning model to interpret the genetically contextual signal of six methylation-sensitive digital PCR (MSdPCR) assays to determine CHD status, with overall area under the curve, sensitivity, and specificity values of 82%, 79% and 76%, respectively. The test uses DNA from whole blood and a relatively inexpensive PCR-based laboratory method; therefore, unlike many of the current diagnostic tests, this new CHD testing approach is inherently more scalable and compatible with the operations of most clinical laboratories. In addition, it presents two distinct potential advantages as compared with current methods for assessing CHD. First, because the MSdPCR assays each measure DNA methylation in pathways involved in the pathogenesis of CHD, it allows clinicians to gather patient specific insight into potentially targetable lifestyle or physiological factors. Second, because DNA methylation is dynamic, it may also be possible to use DNA methylation to serve as a proxy for the success of CHD therapy.

Devising more scalable methods for assessing the success of CHD therapy could have a high clinical impact. The introduction of the hemoglobin A1c (HbA1c) metric for both assessing disease status and monitoring the effects of diabetic therapy revolutionized the treatment of diabetes [8]. However, unlike diabetes, there is not a single pathway whose status can be measured to determine the success of therapy for CHD. As a result, clinicians are relegated to aggregating separate measurements of individual risk factors, such as high blood pressure, elevated cholesterol, or HbA1c status itself, to determine the effectiveness of therapy for CHD. However, many of those measures, such as blood pressure and cholesterol levels, are relatively imprecise, can vary considerably from day to day, and are not sufficient to fully address CHD [9]. Therefore, a more robust test, such as the HbA1c assay, which be used to directly and more precisely assess CHD status and the effectiveness of treatment, would be a significant advancement.

Previously, we have shown that one of the three MSdPCR measures contained in an integrated genetic–epigenetic test for assessing the 3-year risk for CHD significantly changed as a function of treatment for smoking [10]. Since smoking is a key driver of CHD [11,12] and smoking cessation therapy can markedly improve survival in smokers with CHD, this suggests that methylation at one of more of the six loci assessed in the PrecisionCHD test may also respond to smoking therapy. Therefore, in this communication, we assess DNA methylation at each of the six loci in the new CHD test in DNA samples from 39 research subjects collected before and after undergoing 90 days of smoking cessation therapy [13].

## 2. Materials and Methods

The methods and procedures used in this smoking cessation study have been described previously. All methods were approved by the University of Iowa Institutional Review Board (IRB#201706713).

The 39 subjects who participated in this study were part of a 90-day incentive-based smoking cessation program (National Clinical Trials NCT02682147) whose overarching purpose was to understand the relationship between smoking cessation and pulmonary inflammation [14]. Each potential participant, who was alerted to the study by a series of advertisements to patients and staff of the University of Iowa, had to complete a brief online survey in which they reported smoking at least 10 cigarettes per day with a total of five or more pack-years of lifetime consumption in order to qualify for the study. After providing written informed consent, each participant was enrolled in a protocol that included an intake visit and three subsequent monthly follow-up visits at 30-, 60-, and 90 days post intake. At each of these visits, each of the subjects were asked about their current levels of smoking and were phlebotomized to provide DNA for the epigenetic analyses. Due to concerns that medications for smoking cessation may interfere with the pulmonary imaging studies, subjects were instructed not to use medications to quit smoking. However, they were offered USD 400 for having biochemically verified smoking cessation at each of the three follow-up visits. Smoking cessation can take a variable course and self-reports of smoking can be unreliable [15,16]; therefore, smoking intensity at each visit was assessed using digital measures of cg05575921, a generally accepted measure of smoking intensity and smoking cessation [17,18]. At the time of the preparation of the sample, a total of 75 subjects had enrolled in the study, with 39 successfully attending all four clinic visits.

DNA for the epigenetic studies was prepared from whole blood using the method developed by Lahiri and Nurnberger [19]. The MSdPCR assays and bisulfite-sensitive methylation standards used in this study were provided by Cardio Diagnostics Inc. (Chicago, IL, USA) or Behavioral Diagnostics LLC (cg05575921), then used according to the protocols described elsewhere [7,20,21]. In brief, determination of DNA methylation at cg05575921 and the six loci in the PrecisionCHD test (cg03725309, cg12586707, cg04988978, cg17901584, cg21161138, and cg12655112) was conducted using universal droplet digital PCR reagents and equipment obtained from Bio-Rad (Hercules, CA, USA). In brief, 1 µg of DNA was bisulfite-converted using a Qiagen EpiTect Bisulfite kit (Hilden, Germany), according to manufacturer’s directions, with the modified DNA then being eluted in 70 µL of tris buffer. Fourteen cycles of high-stringency PCR amplification of the target region were then performed on a 3 µL aliquot of each bisulfite-converted DNA sample using a set of amplicon-specific proprietary primers. Then, an aliquot of the enriched amplicon target solution was diluted 1:1500, mixed together with primer and probes specific for the target loci and droplet digital PCR reagents, partitioned into droplets with a Bio-Rad droplet generator, and then PCR amplified. Fractional methylation (methylated CpG/(methylated + unmethylation CpG)) of each sample was then determined using a Bio-Rad QX-200 reader and its accompanying Bio-Rad QuantaSoft™ software. The results from all bisulfite methylation testing standards were within their calibrated ranges.

The data were analyzed using the suite of general linear model analytic algorithms embedded in JMP Version 17 (SAS Institute, Cary, SC, USA). Linear regression was used to assess the relationship between each of the six DNA methylation markers in the PrecisionCHD test and the cg05575921 DNA methylation marker for smoking [22]. Comparisons between groups were conducted using Student’s *t*-tests [22]. Bonferroni correction was used to adjust for multiple comparisons [23].

## 3. Results

The clinical and demographic characteristics of the 39 subjects who successfully completed the protocol are given in Table 1. In general, the subjects tended to be White, in their early 40s, and male (22 of 39, or 56%). The subjects tended to be heavy smokers, with the average subject smoking 17 cigarettes per day over the month prior to study intake and having previously had 28 pack-years of total cigarette consumption. The average serum cotinine at intake was 278 ng/mL, with an average Fagerstrom Test for Nicotine Dependence score of 3.7 ± 9.7. Notably, the average cg05575921 methylation value, a generally accepted measure of smoking intensity, was 52.3% at intake.

Table 2 details the identity and chromosomal location of the six CpG sites assessed in the PrecisionCHD test for current CHD assessment, while both Table 2 and Figure 1 delineate the strengths of the linear relationships between methylation at cg05575921 and methylation values at the each of the six CpG loci used in PrecisionCHD at study intake. After correction for multiple comparisons, cg05575921 values were significantly associated with both cg21161138 (Adj R^2^ = 0.91, *p* < 0.0001) and cg12655112 (Adj R^2^ = 0.20, *p* < 0.0048), with a nominal association being further noted with cg03725309 (Adj R^2^ = 0.15, *p* < 0.0162). There was no significant relationship between methylation at cg05575921 and methylation at cg04988978 (Adj R^2^ = 0.04, *p* < 0.21), cg17901584 (Adj R^2^ = 0.09, *p* < 0.07) or cg12586707 (Adj R^2^ = 0.07, *p* < 0.11).

Over the course of the 90-day treatment paradigm, 22 subjects exhibited undetectable levels of cotinine at all three follow-up visits, while the remaining 17 subjects exhibited a positive cotinine value at one or more follow-up visits. Overall, those with negative cotinine levels at all three visits reduced smoking intensity more than those with one or more positive cotinine tests (∆cg05575921; −7.1% ± 6.2 vs. −2.5% ± 5.4; *p* < 0.02), but both groups had decreases in self-reported intake and increases in overall cg05575921 methylation.

Table 3 and Figure 2 illustrate the relationship between the changes in methylation at each of the six loci used in the PrecisionCHD test as a function of cg05575921 methylation. Remarkably, five of the six loci, cg03725309, cg12586707, cg04988978, cg17901584, and cg21161138, exhibited a significant reversion of CHD-associated methylation as a function of cg05575921 which indicated a smoking intensity reduction over the 90-day treatment period. In contrast, there was no significant reversion at cg12655112 (Adj R^2^ = 0.01, *p* < 0.53) at the EH Domain Containing 4 (EHD4) gene CpG site, whose baseline status is associated with smoking intensity.

## 4. Discussion

Creating better, more scalable methods to assess the effectiveness of CHD interventions may be an important step to increasing the effectiveness of CHD treatment. In that regard, our demonstration that five of the six loci of the PrecisionCHD test showing significant changes in response to a 90-day course of smoking cessation is a promising first step. One of the limitations of this study is the relatively small sample size and the cohort being almost exclusively White. Furthermore, we only analyzed one CHD-related intervention: smoking cessation.

The complex dynamics of the methylomic response to successful CHD therapy detailed in this manuscript reflect the complexity of the pathways being interrogated by the test [7]. The six CpG loci targeted by the PrecisionCHD test assess six different molecular pathways moderating vulnerability to CHD. The CpG site in the first pathway, which is assessed by the cg03725309 MSdPCR assay, maps to a candidate cis regulatory element in intron 1 of the seryl-tRNA synthetase 1 gene (SARS1). Demethylation at this locus is associated with obesity, coronary artery calcification, and cardiometabolic syndrome [24,25,26]. The CpG site in the second pathway, measured by the cg12586707 MSdPCR assay, is found in a candidate cis regulatory element approximately 1.5 kb downstream of the 5′ UTR of the CXCL1 gene [27]. CXCL1 is a key member of a group of chemotactic messengers involved in the pathogenesis of a number of inflammatory disorders, such as CHD, and has an important role in the regulation of angiogenesis and cardiac remodeling [27,28]. The CpG site in the third pathway, which is assessed by the cg04988978 MSdPCR assay, maps to the upstream promoter area of myeloperoxidase (MPO), which has been shown to contribute to atherosclerosis by the oxidation of LDL [29]. The CpG site in the fourth pathway, which is assessed by the cg17901584 MSdPCR assay, is found in an intron of 24-Dehydrocholesterol Reductase Divergent Transcript (DHCR24-DT), a long non-coding RNA (lncRNA) gene that is in a divergent (head-to-head) configuration with DHCR24, a key gene in cholesterol biosynthesis [30]. The CpG targeted by the cg21161138 MSdPCR assay maps to the aryl hydrocarbon receptor repressor (AHRR) 267 kb distal to cg05575921 [31]. Similarly to cg05575921, this marker assesses activity in the xenobiotic pathway that is essential for detoxifying the polyaromatic hydrocarbons found in smoke [32]. Finally, the sixth assay, cg12655112, measures methylation at a regulatory region of the first intron of EH Domain Containing 4 (EHD4) gene. EHD4 methylation at this site has been negatively associated with serum glucose levels, while EHD4 expression predicts the success of pancreatic islet transplants [33,34,35].

Both cg05575921 and cg21161138 map to the AHRR gene and both demethylate in response to smoking; therefore, the strong correlation observed between cg05575921 with this new predictor for CHD is to be expected [31]. However, it is interesting to note that the other outright (cg12655112) and nominal (cg03725309) associations with smoking intensity at study baseline were with the two CpG sites whose methylation alterations were previously associated with diabetes [33,36,37]. Smoking is associated with increased risk for diabetes; as such, this makes intuitive sense [6,38]. At the same time, smoking has also been repeatedly implicated as a major cause of inflammation [39]. Thus, the absence of a robust association of smoking intensity with DNA methylation in the inflammatory pathway interrogated by cg12586707 (CXCL1) is somewhat surprising. Then again, it is important to note that the use of the term “inflammation” is often broad, and not all inflammatory processes may be affected by smoking.

To clinicians familiar with the psychosocial processes involved in smoking cessation, the finding that methylation at five of the six pathways interrogated by the MSdPCR assays begin to revert to baseline after only 90 days of therapy is not surprising. In order to cut down or quit smoking, patients often make radical changes to their lifestyle in order to avoid stimuli, such as bars or friends who smoke, that can trigger the urge to smoke. Often, these changes are accompanied with improvements in diet and exercise patterns and decreases in alcohol consumption [40]. Therefore, it should not be surprising that methylation at most, but not all, of the CpG loci targeted in this new diagnostic test were reverted as a function of the reversion of cg05575921 methylation (i.e., successful smoking cessation).

Interestingly, the one locus that did not manifest significant reversion in response to smoking cessation was the EDH4 locus, whose methylation status was associated with baseline cg05575921 methylation. This lack of a significant finding may be secondary to the short time scale of this study. There have been several epigenome-wide studies of the time dependency of methylation reversion in those undergoing smoking cessation [18,41,42]. These studies have demonstrated that the reversion of smoking-associated methylation changes in the thousands of CpG sites affected can be divided into three categories. The first is those whose sites whose methylation status changes quickly, such as cg05575921. The second class is those which revert more slowly, while the third class is those sites which revert very slowly, if at all. Therefore, it should not be surprising to find variation in the time scale of responsiveness at these loci which predict CHD status as well. Given the relationship of cg12655112 (EHD4) methylation with diabetes, it would be interesting to understand if there are any changes in the hemoglobin A1c values of the subjects as a function of smoking cessation in longer-term studies.

Although they are remarkably significant from a statistical point of view, these findings should only be regarded as an initial foray into the epigenetics of effective CHD therapy. Effective treatment of CHD requires targeting the unique lifestyle or physiological factors driving the pathology. For some patients, this means targeting weight; for others, diabetes; for others still, cholesterol levels. However, since successfully addressing these causative factors also often entails broad changes in other clinical drivers of CHD, investigators should be encouraged to use broad measures to fully capture the impact of changes that occur in patients’ lives as a function of successful CHD therapy. Crucially, this was only a 90-day study. Since patients often relapse into unhealthy habits, it is essential to gather longer-term information so that clinicians and public health policy experts can balance the cost of interventions with their long-term impact on CHD outcomes.

Critical to any success to the extension of these findings will be the ability to accurately quantitate changes in the CHD factor being studied. To a large extent, one reason for the success of this study is secondary to the availability of the cg05575921 MSdPCR assay that can precisely measure changes in smoking intensity. Therefore, given the ease of availability of hemoglobin A1c, a precise marker of diabetes status, it would be relatively easy to design a study to test for the reversion of CHD-related signals as a function of changes in hemoglobin A1c values. However, for studies of clinical factors for which less precise steady-state measures are available, such as serum lipid levels, clinicians may need to use innovative strategies or large sample sizes to detect significant changes in methylation. In contrast, the use of self-report variables, such as history of smoking or FTND, which is based solely on self-report and requires a mild degree of introspection, should be used only as necessary because they can be much less reliable.

Avenues through which these findings can be improved upon include the obvious extensions, such as enlarging the sample size and diversity, and more nuanced extensions, such as understanding the role in co-morbid clinical and genetic factors, in moderating reversion. Increases in the size and diversity of the sample are absolutely necessary if this approach is to find clinical application. It is essential to understand how variables such as age, sex, and ethnicity affect reversion before using this to guide clinical care. Furthermore, the subjects in this study were relatively healthy. Understanding the effect of co-morbid illnesses such as obesity or diabetes on clinical response is similarly essential. These examinations should be performed in conjunction with genetic analyses, such as those afforded by polygenic risk scores, which can help parse the acquired effects of any co-morbidity with the inherited genetic factors that contribute to those illnesses.

It may well be that quantifying and understanding these potentially confounding effects is difficult. In this regard, this may be an excellent opportunity for artificial intelligence (AI) to further its role in the derivation and implementation of this testing technology. AI technologies for aggregating and analyzing methylation and genetic data are increasingly commonplace [43,44]. These tools will need to be integrated with tools for parsing clinical laboratory or text data from the electronic medical records in order to create a more complete understanding of the performance and implications of these technologies in the real world [45].

If these trials are adequately powered and treatment compliance is carefully monitored, we believe that additional studies of the methylomics for other types of CHD therapy have a good chance of succeeding. As evidence of that, we note that in a naturalistic longitudinal study of statin therapy, Qin et al. used a mixed linear effects modeling in a sample of 535 Danish subjects to show changes in DNA methylation at three previously identified CpG sites (cg10177197, cg17901584, and cg27243685) as a function of treatment with a variety of statins. Furthermore, using a cross-sectional epigenome-wide approach and data from two large cohorts of subjects, Schrader et al. showed that statin therapy was associated with DNA methylation at one of the sites interrogated by the PrecisionCHD test (cg17901584), as well as two other novel sites (cg27243685 and cg05119988) [46]. Unfortunately, in each of these studies, treatment compliance was not assessed, and both the time and type of statin used varied. However, as evidence of proof of principle, these studies more than demonstrate the potential for precision epigenetic approaches for guiding the treatment of CHD.

In summary, we show significant changes in DNA methylation at five of the six loci measured in the recently described PrecisionCHD test for the assessment of current CHD. We suggest that further studies to understand the methylomic response to CHD treatment may lead to improved methods for guiding the treatment of CHD.

## Figures and Tables

**Figure 1 genes-14-01233-f001:**
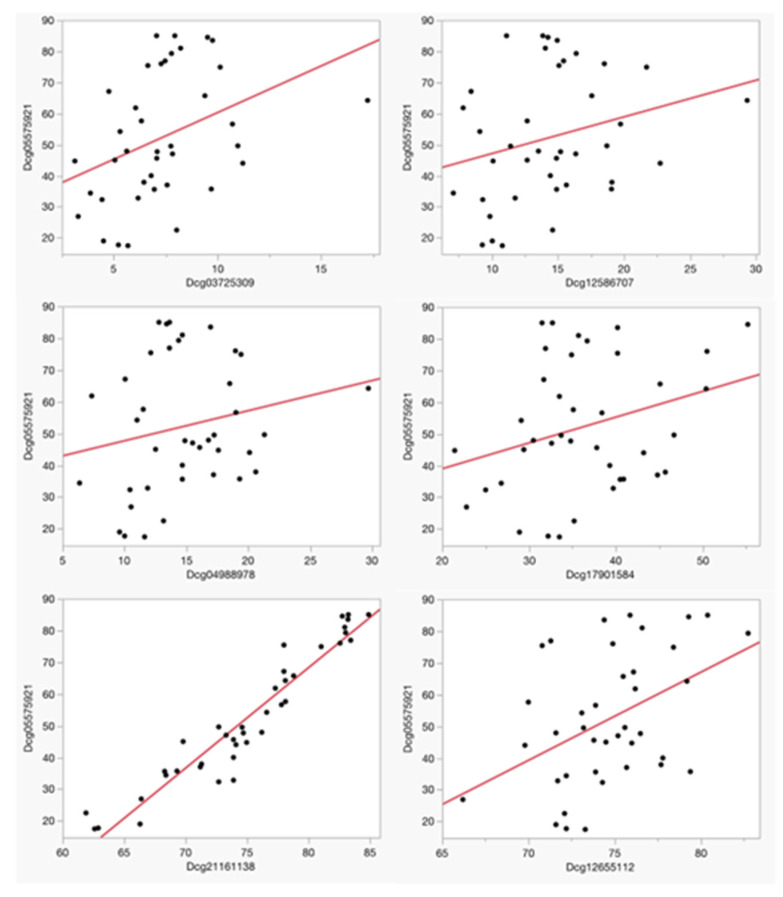
The relationship between baseline Dcg05575921 and the six loci used to assess CHD status. Methylation values are expressed as a percentage. The adjusted R^2^ and *p*-values of the linear associations of each of these comparisons are listed in Table 2.

**Figure 2 genes-14-01233-f002:**
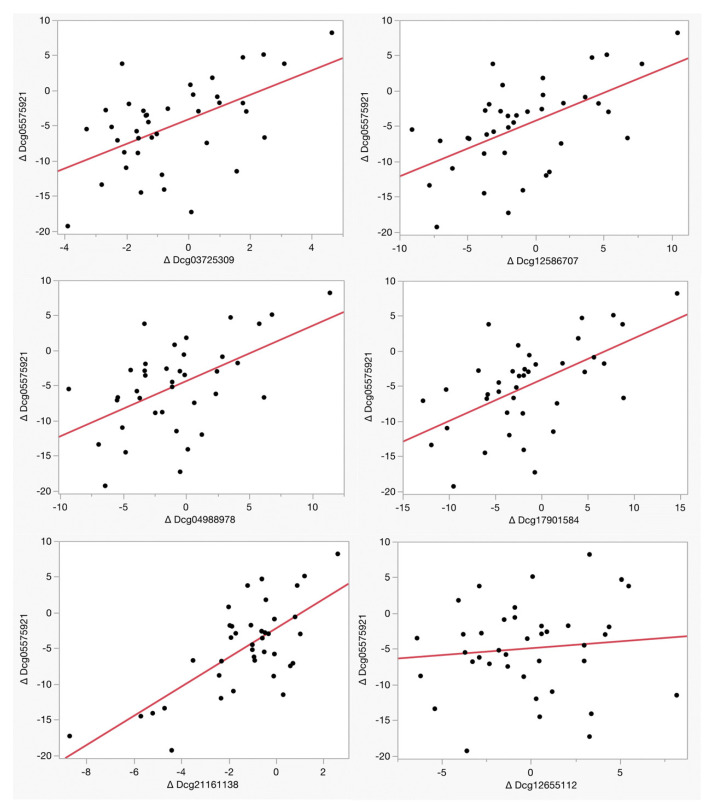
The relationship of the change in smoking intensity with the change in the measure of CHD status. Methylation values for each assay are expressed as percentages. The adjusted R^2^ and *p*-values of the linear associations of each of these comparisons are given in Table 3.

**Table 1 genes-14-01233-t001:** Demographic and clinical characteristics of the subjects.

**N**	39
**Age**	42.7 ± 10.4 years
Gender	
Male	22
Female	17
**Ethnicity**	
White	37
African American	1
Other	1
**Smoking Variables**	
Pack-Year Consumption	28 ± 20
Cigarettes per day	17 ± 9.7
FTND	3.7 ± 9.7
Cotinine (ng/mL)	278 ± 135

FTND, Fagerstrom Test for Nicotine Dependence score.

**Table 2 genes-14-01233-t002:** Methylation values and association with smoking intensity at intake.

Smoking Intensity				
Assay	Gene Localization		Average Value (%)	
Dcg05575921	AHRR		52.3 ± 20.6	
**Coronary Heart Disease (PrecisionCHD)**			**Association with cg05575921**	
			Adj R^2^	Nominal *p*-value
Dcg03725309	SARS1	7.3 ± 2.6	0.12	*p* < 0.02
Dcg12586707	CXCL1	14.4 ± 4.6	0.04	*p* < 0.11
Dcg04988978	MPO	14.8 ± 4.4	0.04	*p* < 0.22
Dcg17901584	DHCR24-DT	36.4 ± 7.6	0.09	*p* < 0.07
Dcg21161138 *	AHRR	72.0 ± 6.2	0.91	*p* < 0.0001
Dcg12655112	EHD4	74.7 ± 3.3	0.19	*p* < 0.005

* *p*-value for significance after multiple corrections is *p* < 0.0083; Dcg is the MSdPCR assay for its respective cg methylation marker.

**Table 3 genes-14-01233-t003:** Linear associations with 90-day changes in cg05575921 methylation.

Smoking Intensity			
Assay		Average Change (%)	
Dcg05575921		−4.9 ± 6.2	
**Coronary Heart Disease (PrecisionCHD)**		Association with ∆cg05575921	
		Adj R^2^	Nominal *p*-value
Dcg03725309	−0.5 ± 1.9	0.29	*p* < 0.0004
Dcg12586707	−0.9 ± 4.5	0.32	*p* < 0.0002
Dcg04988978	−0.7 ± 4.2	0.29	*p* < 0.0004
Dcg17901584	−1.4 ± 6.1	0.33	*p* < 0.0001
Dcg21161138	−1.3 ± 2.1	0.50	*p* < 0.0001
Dcg12655112	−0.1 ± 3.4	0.01	*p* < 0.53

*p*-value for significance after multiple corrections is *p* < 0.0083; Dcg is the MSdPCR assay for its respective cg methylation marker.

## Data Availability

The datasets used during the current study are available from the corresponding author on reasonable request.

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
