# Peer review of "The Reversion of the Epigenetic Signature of Coronary Heart Disease in Response to Smoking Cessation"

_genes, 2023, doi:10.3390/genes14061233_

Round 1
Reviewer 1 Report
the paper is good and well written
Reviewer 2 Report
Dear Authors,
Your study is well presented and innovative, yet I believe it is not very interesting in the clinical practice.
In the methods section, it is not clearly stated how the compliance to the stop- smoking habit has been verified. In the description of the study population, CHD risk factors of the study partecipants should be described and related to the genetic results. Moreover, the cost of the analysis proposed is not specified and related to that of conventional CHD follow up methods.
I think you should improve your manuscript and discuss more critically the clinical applications of your study.
Best regards
I think minor English revision is required
Reviewer 3 Report
The present study brings an interesting relationship between smoke habits and gene activation.
Although the physiologic mechanisms and biochemistry overcomes tease scientific curiosity, the objectives of investigation were clear presented.
The methylation is a mechanism the regulates genes expression that must be explored, and relational triggers that can act on it.
The authors relate a high-risk death acute illness with habits and genetic conditions that be changed.
The pathogenesis is the most secure path to understand the multifactorial context. These information guides for different types of approaches.
This study analyzed a hypothetical relationship methylation change in six different loci of gene cg055759211 using a 39 smoke quit individuals. The potential application can guide clinicians to categorized patients in coronary heart disease.
Nevertheless, the sample is not enough representative to take convicted results.
The statistical were developed enough to justify correlation of methylation state of gene.
The discussion is precisely written to ground the perspectives of first concept epigenetic mechanism of CHD pathogenesis.
In overall aspects the study is a part of large epigenetic study which can brings more answers of gene based pathogenesis.
Major points – review statistic assays presentation including a conserved methylation gene as control to reinforce smoke quite the mechanism.
- Enlarge samples (obvious is possible);
- Discuss individual profile in response of smoke cessation and genes methylation.
Minor points
- present the literature review about the artificial intelligence for gene detection.
- Present the FTND importance for smokers ( the SD is quite high – 3.7+/-9.7)
Reviewer 4 Report
The manuscript is very interesting and provides good insight into the reversibility of epigenetic changes through lifestyle modification. In addition, this manuscript provides the opportunity to apply new diagnostic tools, such as determining the methylation status of specific genes in monitoring recovery from certain diseases.
The manuscript is well and clearly written and adequately presents the results and discussion.
However, the authors should clarify some things in the manuscript:
Lines 119-120 - it is written that DNA was isolated from whole blood by cold protein precipitation as described in Ref. 19. However, Ref. 19 refers to obtaining high molecular weight DNA by salting out cellular proteins by dehydration and precipitation with saturated sodium chloride solution.
Figure 1 is a bit blurry, unlike Figure 2, which is decently sharp.
Line 298 - 'those sites whose methylation status changes quickly' should be revised because it needs to be written clearly.
Minor editing of English language required.
Round 2
Reviewer 2 Report
I am satisfied with the authors ' reply to my review
Minor spell Check is needed